# Effect of Various Intermittent Fasting Protocols on Hyperglycemia-Induced Cognitive Dysfunction in Rats

**DOI:** 10.3390/brainsci13020165

**Published:** 2023-01-18

**Authors:** Hani A. Alfheeaid, Ahmad A. Alhowail, Faiyaz Ahmed, Abdel Kader A. Zaki, Areej Alkhaldy

**Affiliations:** 1Department of Food Science and Human Nutrition, College of Agriculture and Veterinary Medicine, Qassim University, Buraydah 51452, Saudi Arabia; 2Department of Pharmacology and Toxicology, College of Pharmacy, Qassim University, Buraydah 51452, Saudi Arabia; 3Department of Clinical Nutrition, College of Applied Health Sciences in Ar Rass, Qassim University, Al Qassim Region, Ar Rass 51921, Saudi Arabia; 4Department of Veterinary Medicine, College of Agriculture and Veterinary Medicine, Qassim University, Buraydah 51452, Saudi Arabia; 5Department of Physiology, Faculty of Veterinary Medicine, Cairo University, Giza 12211, Egypt; 6Clinical Nutrition Department, Faculty of Applied Medical Sciences, King Abdulaziz University, Jeddah 21589, Saudi Arabia

**Keywords:** animal model, diabetes mellitus, diet, cognitive impairment, hyperglycemia, energy restriction, fasting

## Abstract

Diabetes mellitus is a highly prevalent metabolic disorder that causes cognitive decline. Here, we investigated the impact of various intermittent fasting protocols on type 2 diabetes mellitus (T2DM)-induced cognitive dysfunction in a rodent model. Male Sprague–Dawley rats (aged 3 months) were randomly assigned to five groups (n = 6 per group) and T2DM was induced by streptozotocin (60 mg/kg, IM). The control group was untreated. Cognitive function was tested (Y-maze, novel object recognition, and elevated plus maze tests) and glucose was assessed. The T2DM rats exhibited significantly higher blood glucose, which is associated with cognitive dysfunction. Compared to the validated animal model of T2DM in rats, various intermittent fasting protocols decreased blood glucose and improved cognitive function. These results indicate that various intermittent fasting protocols may be a potential strategy for managing the hyperglycemia-associated cognitive dysfunction.

## 1. Introduction

Type-2 diabetes mellitus (T2DM) is a metabolic disorder that results from insulin depletion and/or failure of tissues to respond to insulin (insulin resistance) and metabolize glucose, which ultimately results in chronic hyperglycemia [1]. T2DM is known to disrupt carbohydrate, lipid, and protein metabolism and have severe consequences, including long-term complications in many vital organs [2].

The global incidence of T2DM is drastically escalating, and its prevalence is estimated to reach more than 200 million by 2035 (International Diabetes Federation, Diabetes Atlas 2015). The progressive effects of T2DM extend beyond the disease manifestation itself, with complications that affect the structure and function of other physiological systems and organs, resulting in cerebrovascular dysfunction, renal failure, visual impairment, sexual dysfunction, and dementia [3,4]. Insulin plays a vital role in learning and cognition by significantly affecting the central and peripheral downstream signaling pathways [5,6]. The well-known and accepted neurosignaling mechanism is insulin binding to its cognate receptor, which causes phosphorylation of insulin receptor substrates 1 (IRS-1) [7], leading to the activation of IRS-1 that results in downstream phosphorylation of phosphatidylinositol 3-kinase (PI3K), and protein kinase B (PkB/Akt) [8]. This inactivates glycogen synthase kinase-3 beta (GSK3β) by phosphorylating Ser-9 residue, thereby enhancing learning and cognitive processes [9]. In contrast, long time of hyperglycemia can damage the blood vessels in the brain, which reduces carrying oxygenated blood within the brain regions leading to cognitive impairment and vascular dementia [10,11]. Therefore, there are several natural and synthetic drugs that can decrease the hyperglycemic effects associated with T2DM. However, there are very few drugs such as metformin that can ameliorate the cognitive dysfunction associated with hyperglycemia [12].

Intermittent fasting (IF) is a form of calorie-restriction (dieting) treatment for managing body weight [13]. Compared to typical calorie-restriction diets, IF provides a more flexible approach to reduce total caloric intake by limiting time/meal frequency [14]. Individuals following IF protocols are allowed to eat at certain times or days and fast during others. The IF protocol is typically presented in three forms: (1) alternate-day fasting, (2) time-restricted fasting, and (3) periodic fasting [15]. Such dieting patterns produce time variations in consecutive numbers of fasting hours or days between feeding periods [15]. Previous strategy of IF protocols did not permit any dietary intake during fasting days. However, this strategy is not reasonable for maintaining good compliance with the diet and avoiding the health risks of prolonged fasting in weight management programs. Thus, current strategy of IF protocols (modified IF) allow some energy intake during the fasting days [15,16]. Overall, current IF protocols generally aim to limit daily energy intake up to 50 % of total energy requirements [15].

Despite of the clinical reports on the beneficial effect of various intermittent fasting protocols on hyperglycemia and cognitive function. There are limited preclinical studies supporting this notion. The current study provides experimental evidence of the effect of various intermittent fasting protocols on hyperglycemic control and associated with cognitive impairment.

## 2. Material and Methods

### 2.1. Chemicals

The chemical streptozotocin (STZ) was purchased from Cayman Chemical (Ann Arbor, MI, USA).

### 2.2. Treatment Protocol

The experimental protocols used in the present study were approved by the Institutional Animal Ethics Committee, and the Deanship of Scientific Research, and College of Pharmacy, Qassim University, Saudi Arabia [Approval ID 2020-CP-12] 10137-cavm-2020-1-3-I. Male Sprague–Dawley rats (aged 3 months) were divided into five groups (six rats per group): a control group (NC), diabetes control group (DC), continuous calorie restriction group (CCR), alternate day calorie restriction group (ALT), and a calorie restriction for 2 days group (PF). The control group received an intramuscular (IM) injection of a vehicle (0.5 % w/v carboxyl methyl cellulose sodium). Hyperglycemia was induced in the other groups by an IM injection of STZ (55 mg/kg). The development of hyperglycemia was confirmed by measuring the rats’ blood glucose levels (Accu-Chek Glucometer, Roche, Germany) after 72 h and on day 4 after the last injection. Rats with fasting blood glucose levels higher than 126 mg/dl were considered to be diabetic and used as subjects in further studies [17]. The four fasting protocols were started after six weeks (Figure 1).

### 2.3. General Behavioral Assessment of Rats

The health and condition of the animals was monitored every day. All the rats were subjected then to the behavioral tests and had their glucose levels measured.

### 2.4. Y-Maze Test

The Y-maze is a behavioral test used to assess the ability of an animal to recognize familiar places and its tendency to explore new places [18]. In this study, Y-maze tests were used to measure spatial memory in the control and experimental animals. The apparatus, which consisted of three wooden arms at 120-degree angles to each other, was placed on the floor with a light at the top of the maze to distribute light evenly. A camera was placed above the maze to record the test sessions. The Y-maze tests (each rat was tested individually) were conducted on day 42 after the hyperglycemia induction. During a 10-min training session, the rats were allowed to explore freely only two of the arms. The arm in which they were initially placed (start arm) and one other (familiar arm). During the second session (the test session), the rats were allowed to explore the entire maze, including the newly available arm (novel arm), for 3 min. The interval between the first and second sessions was 3 h. The second session was videotaped and the number and order of entries into the novel arm, and the dwell time spent in the novel arm, were recorded for each rat.

### 2.5. Novel Object Recognition (NOR) Test

The NOR test is a behavioral test to measure hippocampus-dependent memory in experimental animals [19]. The apparatus consisted of a wooden box (40 × 40 × 40 cm) with an open top and a camera placed above to record the test sessions. In this test, each rat was introduced to identical objects and allowed to explore them for 10 min. After 3 h, one of the objects was replaced before the rat was returned to the box and allowed to explore the objects. The dwell time spent exploring the new object was recorded [20].

### 2.6. Elevated plus Maze (EPM) Test

The EPM is a behavioral test used to measure cognitive function in experimental animals [21]. The apparatus consisted of four arms at 90-degree angles to each other. Two arms were open and two were covered. Each arm was 50 cm in length and 10 cm wide, and the height of the sidewalls was 30 cm. There was an open central area of the maze that measured 10 cm^2^. The apparatus was placed 50 cm above the floor, and a camera was placed above it to record the test sessions. In the training session, each rat was placed at the end of an open arm, facing away from the central platform, and allowed to explore the apparatus for 5 min. After 3 h, the rat was returned to the maze and transfer latency (TL) was recorded as the time for the rat to move from the end of the open arm to its entry into a closed arm. The camera recoded the rats’ behavior for analysis [22].

## 3. Results

### 3.1. Fasting Protocols Did Not Improve the Body Weight in the Diabetic Rats

The body weights of rats in the diabetic rats were significantly reduced compared to the control group. However, the various intermittent fasting protocols groups on diabetic rats were not altered compared with those in the diabetic groups and still reduced compared to the control group (*p* < 0.001) (Figure 2).

### 3.2. Fasting Protocols Reduced Some of the Hyperglycemia-Induced Cognitive Dysfunction as Measured by the Y-Maze Test

Compared with the control group, the number of entries and the time spent in the novel arm were significantly lower in the diabetic control rats, which indicated cognitive impairment (Figure 3 and Figure 4, respectively, *p* < 0.05 and *p* < 0.001). However, the hyperglycemia-induced cognitive impairment was reduced in a dose-dependent manner across all the fasting protocols used in this study (as measured by the number of entries). However, the duration of time spent in the novel arm did not increase in the calorie restriction for 2 days group.

### 3.3. Fasting Protocols Reduced Some of the Hyperglycemia-Induced Cognitive Dysfunction as Measured by the NOR Test

Compared with the control group, the diabetic rats (DC), and in the calorie restriction for 2 days (PF) groups spent significantly less time exploring the novel object (*p* < 0.001). Similarly, the continuous calorie restriction (CCR) and alternate day calorie restriction (ALT) groups spent significantly less time exploring the novel object (*p* < 0.01) (*p* < 0.05), respectively. These results indicate impairments in hippocampus-dependent cognitive function in the diabetic rats was reduced in CCR and ALT by mitigating the effect of diabetes (compared to the control group), as measured by time spent exploring a novel object (Figure 5).

### 3.4. Fasting Protocols Reduced Some of the Hyperglycemia-Induced Cognitive Dysfunction as Measured by the EPM Test

Compared with the control group, the diabetes control group and the continuous calorie restriction group had longer transfer latency times in the EPM test. However, the alternate day calorie restriction group and the calorie restriction for 2 days group had transfer latency times that were only slightly higher than the control group and lower than the diabetes control and continuous calorie restriction groups (Figure 6).

### 3.5. Fasting Protocols Reduced the Elevated Glucose Levels in the Hyperglycemic Rodent Model

Blood glucose levels were measured on days 1, 7, 14, 21, and 28. Diabetic rats exhibited significantly higher glucose content in their blood, compared to the control group (Figure 7, *p* < 0.001). All fasting protocols significantly reduced the levels of glucose compared to the diabetic control group. The calorie restriction for 2 days group (*p* < 0.01) and the alternate day calorie restriction group (*p* < 0.001) had significantly lower glucose levels on day 28 compared to the diabetic control group (Figure 7).

## 4. Discussion

Diabetes mellitus is one of the fastest growing disorders worldwide, and it has high morbidity and mortality rates [23]. One of the complications of diabetes mellitus is cognitive impairment [24] and several lines of evidence have revealed that insulin resistance is one of the major mechanisms that induce cognitive impairment [25,26]. Furthermore, there is evidence of a link between prolonged abnormalities in blood glucose levels (hyperglycemia), lack of insulin content, and insulin resistance in the brain in diabetes-mellitus that is related to cognitive dysfunction [27]. Fasting is one of the non-pharmacological strategies used to control T2DM. However, there are few studies that have demonstrated the neuroprotective effects of fasting protocols against hyperglycemia-induced cognitive impairment [28,29,30]. Thus, the present study investigated the neuroprotective effects of various intermittent fasting protocols against hyperglycemia-induced cognitive dysfunction in rats. Moreover, this study evaluated behavioral measures (the Y-maze, novel object recognition, and elevated maze tests) and glucose levels following various intermittent fasting protocols in a scientifically accepted animal model of hyperglycemia.

Using the Y-maze, NOR, and EPM behavioral tests, we demonstrated impaired cognition in T2DM rats, which is consistent with previous studies [31,32,33]. The Y-maze test of spatial memory revealed a decrease in the number of entries into the novel arm and the dwell time in the novel arm in T2DM animals, indicating they had cognitive impairment, compared to the control group (Figure 2 and Figure 3). Rats that followed various intermittent fasting protocols (the CCR, ALT, and PF groups) showed improved cognitive function, as measured by higher numbers of entries into the Y-maze. However, only the CCR and ALT groups showed improved cognitive function, as measured by longer dwell times in the novel arm. In addition, the NOR test revealed that all four groups of T2DM animals spent significantly less time exploring a novel object than the control group did (*p* < 0.001). Whereas the CCR and ALT groups had higher exploration times than the diabetic control group (*p* < 0.01, *p* < 0.05, respectively), the exploration time of the PF group did not differ from that of the diabetic control group. Furthermore, T2DM animals had higher transfer latency times, compared to the control animals. However, some of the intermittent fasting groups showed reductions (the ALT and PF groups) and there was no difference between the CCR group and the diabetic control group. Altogether, various intermittent fasting protocols can generally restore the loss of cognitive function caused by diabetes.

Insulin and insulin receptors play an essential role in regulating brain function, as well as being involved in learning, cognitive processes, and synaptic plasticity [34]. Insulin receptors are highly expressed in neurons, particularly in the hippocampus [35,36]. Insulin signaling plays an important role in neuronal survival and synaptic plasticity [37]. Insulin administration has been shown to improve learning and cognitive processes significantly in patients with T2DM [38,39]. Insulin binds to insulin receptors to activate the PI3K/Akt signaling pathway [40], which plays a vital role in glucose transporter 4 (GLUT4) traveling to the cell surface that ultimately reduces hyperglycemia [41]. In this study, we used a rat model of T2DM induced by STZ that caused hyperglycemia and induced cognitive impairment in animals. It is reported that long time hyperglycemia could damage the blood vessels in the brain affecting the blood delivery to the brain regions that leads to ischemia and impairment of the cognitive function [10,11]. Interestingly, our data revealed that various intermittent fasting protocols of diabetic rats improved behavioral performance and glucose levels, comparable to those in the control group.

There are some limitations to this study. This study is conducted on the experimental animal rat model of diabetes, which is induced their diabetes by utilizing STZ. These wildtype rats used to assess the effects of various intermittent fasting protocols on cognitive function and the body weight without evidence on its effect on insulin sensitivity or receptor signaling. Therefore, it is worth mentioning the effect of various intermittent fasting protocols on insulin sensitivity or receptor signaling as well as metabolic health. Although the result indicated some of these various intermittent fasting protocols improve the cognitive function and seems reducing the cytotoxicity, the mechanisms of these improvements in the cognitive function could be a future area and the direction where further research is required be as well as it is applied on diabetic patients.

In conclusion, our study showed that diabetes impairs cognitive function and various fasting protocol restored memory function that was reduced by diabetes, and improved behavioral performance on the Y-maze, NOR, and EPM tasks. Similarly, although the glucose levels of the rats were increased by diabetes, various fasting protocols significantly reduced their glucose levels. Conclusively, various fasting protocols can be important for helping patients with diabetes and practitioners in their selection of the most affective fasting protocol to manage glucose levels.

## Figures and Tables

**Figure 1 brainsci-13-00165-f001:**
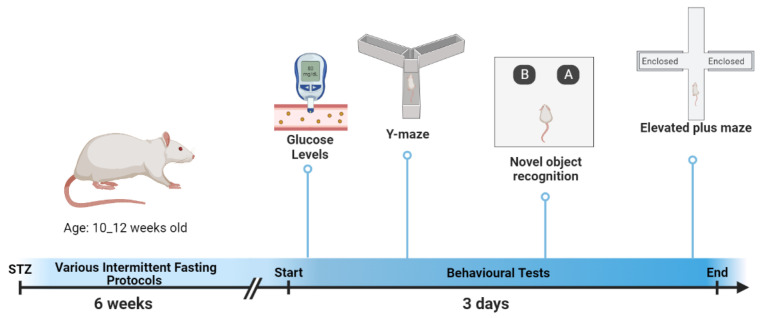
Conceptual diagram explaining the timeline of the study starting from the diabetes induction to conducting all experiments.

**Figure 2 brainsci-13-00165-f002:**
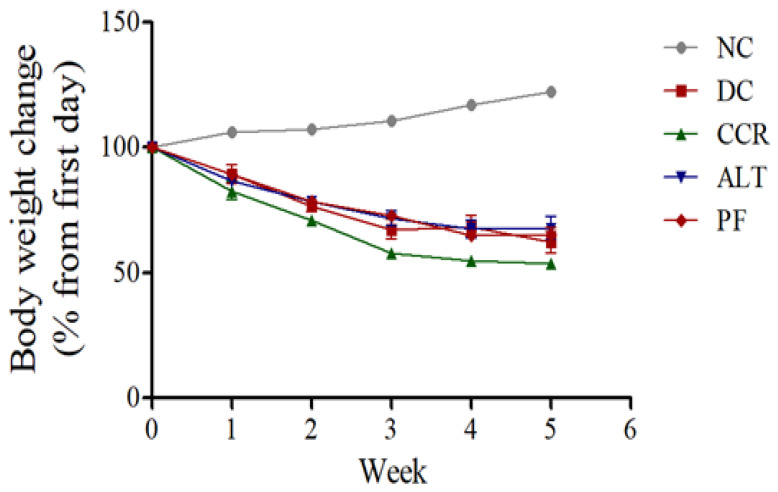
Effects of various intermittent fasting protocols on rat body weight. The graphs show the average body weight per group from the first day of the fasting protocols to the end of the study. The data revealed no significant changes in body weight during the study period in all diabetic groups. However, there was a significant difference in all of the diabetic versus control groups (*p* < 0.01). Data are expressed as the mean ± standard error of the mean. NC = Control, DC = Diabetes, CCR = Continuous calorie restriction, ALT = Alternate day calorie restriction, PF = Calorie restriction for 2 days.

**Figure 3 brainsci-13-00165-f003:**
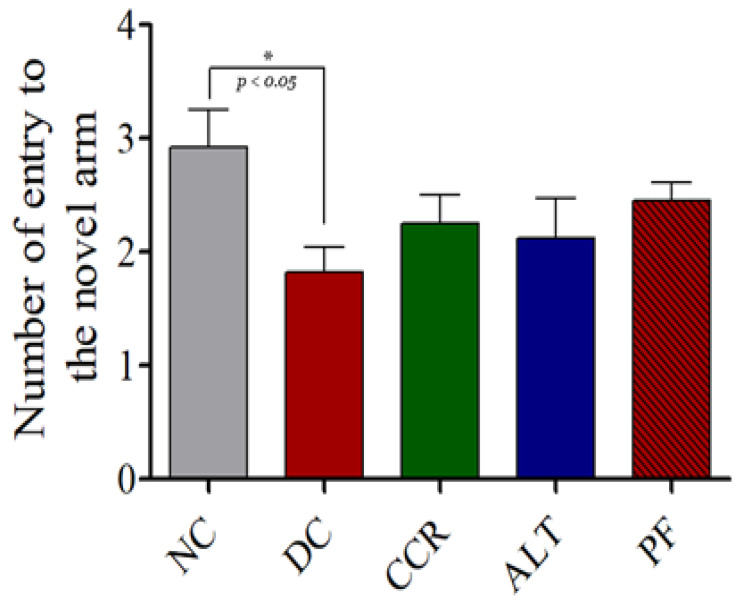
Effects of various intermittent fasting protocols on the number of entries into the novel arm of the Y-maze. The results revealed that the number of entries into the novel arm were significantly lower in the DC group compared to the control group, but not significantly lower in the CCR, ALT, or PF groups compared to the control group (* *p* < 0.05). Data are expressed as the mean ± standard error of the mean. NC = Control, DC = Diabetes, CCR = Continuous calorie restriction, ALT = Alternate day calorie restriction, PF = Calorie restriction for 2 days.

**Figure 4 brainsci-13-00165-f004:**
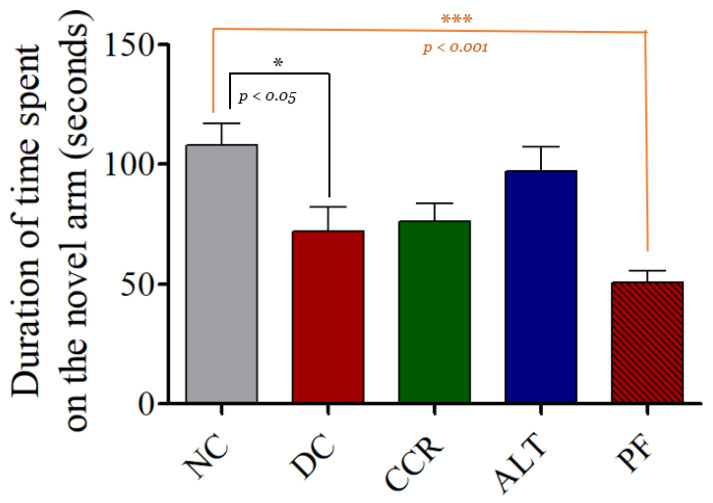
Effects of various intermittent fasting protocols on the total time spent in the novel arm of the Y-maze. The results revealed that the time spent in the novel arm was significantly shorter in DC and PF groups. However, the CCR and ALT groups did not differ significantly from the control group (* *p* < 0.05 and *** *p* < 0.001 vs. control group). Data are expressed as the mean ± standard error of the mean. NC = Control, DC = Diabetes, CCR = Continuous calorie restriction, ALT = Alternate day calorie restriction, PF = Calorie restriction for 2 days.

**Figure 5 brainsci-13-00165-f005:**
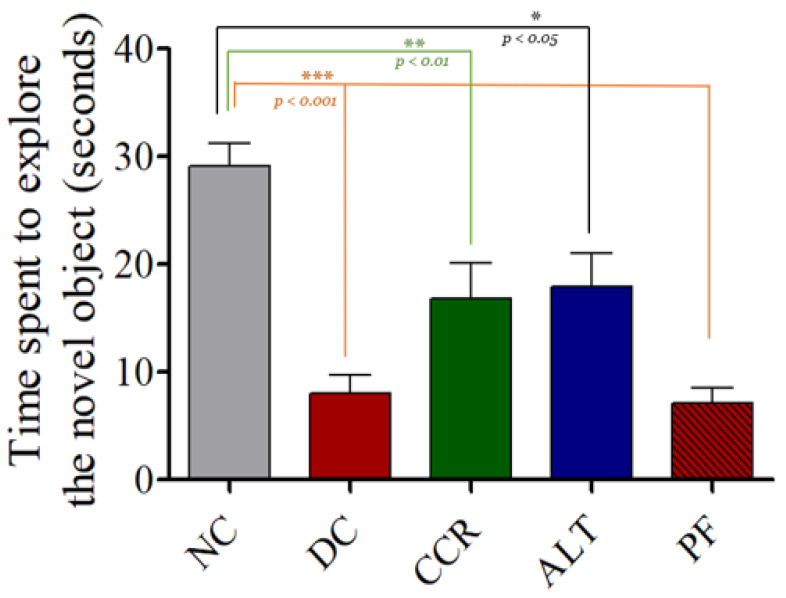
Effects of various fasting protocols on cognitive function using novel object recognition. Data represented as the mean ± standard error of the mean (SEM); * indicates significant difference between the control group and the various fasting protocols (* *p* < 0.05, ** *p* < 0.01 and *** *p* < 0.001 vs. control group): DC and PF (*p* < 0.001), CCR (*p* < 0.01), ALT (*p* < 0.05); results of one-way ANOVA. NC = Control, DC = Diabetes, CCR = Continuous calorie restriction, ALT = Alternate day calorie restriction, PF = Calorie restriction for 2 days.

**Figure 6 brainsci-13-00165-f006:**
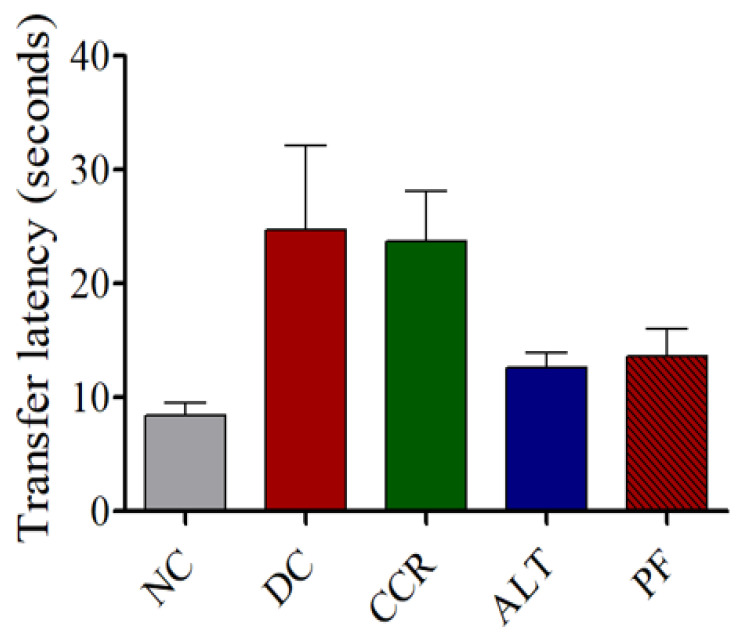
Effects various intermittent fasting protocols on cognitive behavioral performance of control and hyperglycemic rats in the elevated plus-maze test. Data represent the mean ± SEM of the transfer latency time. NC = Control, DC = Diabetes, CCR = Continuous calorie restriction, ALT = Alternate day calorie restriction, PF = Calorie restriction for 2 days.

**Figure 7 brainsci-13-00165-f007:**
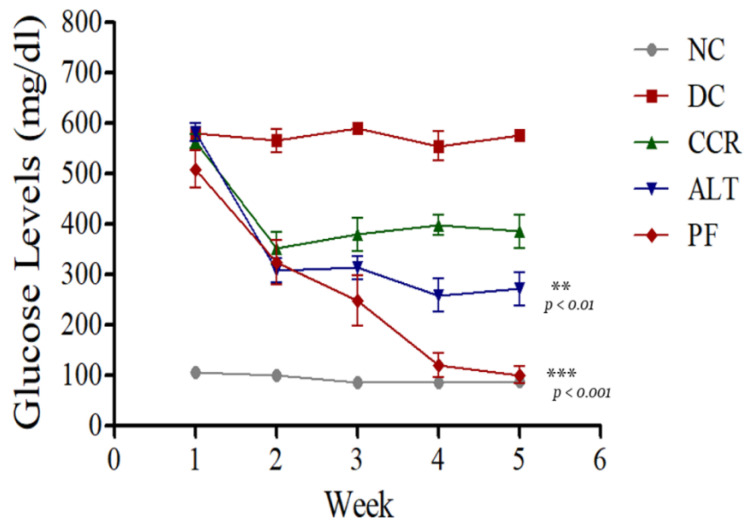
Effects of various intermittent fasting protocols on glucose levels: Data represent the mean ± SEM of the glucose content (** *p* < 0.01 and *** *p* < 0.001 vs. diabetic group). NC = Control, DC = Diabetes, CCR = Continuous calorie restriction, ALT = Alternate day calorie restriction, PF = Calorie restriction for 2 days.

## Data Availability

Not applicable.

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
