# Peer review of "Effect of Various Intermittent Fasting Protocols on Hyperglycemia-Induced Cognitive Dysfunction in Rats"

_brainsci, 2023, doi:10.3390/brainsci13020165_

Round 1
Reviewer 1 Report
Comments and Suggestions for Authors
Review
Title: Effect of Various Intermittent Fasting Protocols on Hyperglycemia-induced Cognitive Dysfunction in Rats.
Dear Author,
Thank you for this interesting study. I think that the study reflects the effect of glycemic levels on brain cognitive function a very important aspect of diabetes that is rarely assessed. Assessing human brain cognitive function is very challenging and not straight-forward. However, I think is that the study was performed on rats and to apply the findings to human is a bit controversial. Another point to consider is that glycemic effects on brain function occurs over long periods of time could be more than 42 days, I think longer periods and more frequent follow-ups should be considered in further analysis. Another suggestion, I think measuring the difference between brain cognitive function before and after intermittent fasting for the same rat will give better reflection of the effect of glycemic control on brain function. The introduction explains the effect of insulin level on cognitive brain function, I wonder has the insulin levels were tested in the study. The overall study is very valuable and uncovers important area of diabetes.
The following are few minor suggestions about the manuscript formatting.
Suggestions to the author
Lines 36-51 The study focuses on measuring of glucose level and relating it to the cognitive function of the brain. This paragraph describes the effect of insulin on the brain cognitive function and not glucose level. I suggest relating signaling pathways to glucose levels.
Line 51 “there are very few drugs that can ameliorate the cognitive dysfunction associated with hyperglycemia”, I suggest adding examples of drugs and suitable reference(s).
Lines 52-64 I noticed that only one reference is added to the line, I suggest adding more references about intermittent fasting and glycemic control.
Discussion I suggest adding more information about how the brain function is affected by hyperglycemia, how cognitive functions decline supported by suitable references.
Figures. I think figures and their legends should be self-explanatory. I suggest adding the legend abbreviations in full words in the figure caption.
Author Response
Review 1
|
Comments |
Responses |
1 |
Lines 36-51 The study focuses on measuring of glucose level and relating it to the cognitive function of the brain. This paragraph describes the effect of insulin on the brain cognitive function and not glucose level. I suggest relating signaling pathways to glucose levels.
|
Added |
2 |
Line 51 “there are very few drugs that can ameliorate the cognitive dysfunction associated with hyperglycemia”, I suggest adding examples of drugs and suitable reference(s).
|
Added |
3 |
Lines 52-64 I noticed that only one reference is added to the line, I suggest adding more references about intermittent fasting and glycemic control.
|
added |
4 |
Discussion I suggest adding more information about how the brain function is affected by hyperglycemia, how cognitive functions decline supported by suitable references.
|
Added |
5 |
Figures. I think figures and their legends should be self-explanatory. I suggest adding the legend abbreviations in full words in the figure caption.
|
Corrected |
Reviewer 2 Report
Comments and Suggestions for Authors
The type of paper should be changed from "Article" to "Communication"
Limits, advantages and further directions of research should be added.
Lines 52-62 should be better commented.
A graphical scheme of study design should be added.
Data in Figure 1 should be better described.
Results on NOR test should be better described.
Author Response
Review 2
|
Comments |
Responses |
1 |
The type of paper should be changed from "Article" to "Communication"
|
Needed to be published as "Article" |
2 |
Limits, advantages and further directions of research should be added.
|
added |
3 |
Lines 52-62 should be better commented. |
Done |
4 |
A graphical scheme of study design should be added.
|
Added |
5 |
Data in Figure 1 should be better described.
|
Done |
6 |
Results on NOR test should be better described. |
Done |